# FEW-SHOT CLASSIFICATION ON GRAPHS WITH STRUCTURAL REGULARIZED GCNS

## ABSTRACT

We consider the fundamental problem of semi-supervised node classification in attributed graphs with a focus on *few-shot* learning. Here, we propose Structural Regularized Graph Convolutional Networks (SRGCN), novel neural network architectures extending the well-known GCN structures by stacking transposed graph convolutional layers for reconstruction of input features. We add a reconstruction error term in the loss function as a regularizer. Unlike standard regularization such as $L_1$ or $L_2$, which controls the model complexity by including a penalty term depends solely on parameters, our regularization function is parameterized by a trainable neural network whose structure depends on the topology of the underlying graph. The new approach effectively addresses the shortcomings of previous graph convolution-based techniques for learning classifiers in the few-shot regime and significantly improves generalization performance over original GCNs when the number of labeled samples is insufficient. Experimental studies on three challenging benchmarks demonstrate that the proposed approach has matched state-of-the-art results and can improve classification accuracies by a notable margin when there are very few examples from each class.

## 1 INTRODUCTION

Graph is the most natural structure to model interactions between different entities. For example, graphs are often used to model social networks (Facebook, Twitter), biological networks (protein-protein interaction), and citation networks (arXiv). There has been an increasing research interest in learning tasks on arbitrary structured graphs recently, e.g., (Bruna et al., 2014; Monti et al., 2017; Defferrard et al., 2016; Kipf & Welling, 2017; Hamilton et al., 2017; Velickovic et al., 2018).

In this paper, we consider the fundamental problem of semi-supervised node classification in attributed graphs, where the labels of a small training set are available. Such label information is critical to the performance of classification algorithms. However, it usually requires the inspection of human experts, and thus is often expensive to obtain sufficient label information. It is therefore important to design effective learning methods for the setting where only minimum amount of supervision is available: the classifier needs to generalize well after seeing few or even zero training samples from each class.

To tackle the complexity of huge real-world networks, representation learning-based approaches have become popular for learning tasks on structured data. One line of research is *network embedding* (Tang et al., 2015; Grover & Leskovec, 2016; Bojchevski & Günnemann, 2018). This approach first learns a lower-dimensional representation for each node in the graph in an unsupervised manner, followed by a supervised classifier for node classification, e.g., logistic regression or multi-layer perceptron (MLP).

Another popular line of research is trying to generalize convolutional neural networks (CNNs) to graph data, e.g., (Bruna et al., 2014; Duvenaud et al., 2015). One family of such methods that is most related to the present work is Graph Convolutional Networks (GCNs) Kipf & Welling (2017). GCNs are neural network models operating directly on graphs, which can be viewed as an effective first-order approximation of spectral graph convolutions Hammond et al. (2011).

The above two approaches achieve impressive classification accuracies and have drawn a great amount of attention. However, deep neural networks generally perform poorly on few-shot learn-

ing tasks Ravi & Larochelle (2017). This work proposes a simple yet powerful framework, called Structural Regularized Graph Convolutional Network (SRGCN), for few-shot and zero-shot learning tasks on graphs.

In the conventional GCN model of Kipf & Welling (2017), the convolution filter is restricted to operate in a 1-hop neighborhood. In each layer, the hidden features are first linearly transformed by a trainable weight matrix; the new hidden feature vector of a node is computed by aggregating the hidden features from its neighboring nodes followed by a non-linear activation. Such iterative aggregation schemes have proven to effectively learn node representations and achieve state-of-the-art classification accuracy by utilizing topological and feature information simultaneously. When training examples are abundant, label information can be efficiently propagated to the entire graph in a couple of iterations; indeed, the best performance of GCN is typically achieved with a 2-layer architecture Kipf & Welling (2017). On the other hand, when labels are extremely scarce, intuitively more rounds of aggregations are needed hence deeper networks; a theoretical analysis on the limitation of shallow GCN for few-show learning is presented in Section 3. However, deeper versions of GCN often lead to worse performance Kipf & Welling (2017); Xu et al. (2018), and according to the analysis of Xu et al. (2018), this is mainly because some critical feature information may be "washed out" via the iterative averaging process.

**Present Work.** To resolve this dilemma, we propose a novel regularized graph convolutional network architecture, called SRGCN. Regularization is usually applied to prevent over-fitting Kipf & Welling (2017); Velickovic et al. (2018), however, the effect of standard regularization methods, which controls the model complexity by including a penalty term depends solely on the parameters, is limited in few-shot learning on graphs. Our regularization on the other hand crucially depends on the structure of the underlying graph, whose form itself is also trainable. We write the loss function as $\mathcal{L} = \mathcal{L}_{sup} + \alpha \mathcal{L}_{sr}$, where $\mathcal{L}_{sup}$ is the supervised loss and the structural regularization term $\mathcal{L}_{sr}$ is:

$$\mathcal{L}_{sr} = \|g_{\theta_2} \circ f_{\theta_1}(X) - X\|^2.$$

Here $\circ$ denotes function composition, $X$ is the input features, $f_{\theta_1}(X)$ is the encoding function defined by a usual GCN, and $g_{\theta_2}()$ is a reconstruction function parameterized by a transposed graph convolutional network. For fixed $X$ and $\theta_2$, $\mathcal{L}_{sr}$ is a function of the GCN parameter $\theta_1$ that we want to learn, hence it is indeed a regularizer for GCN. Note this encoder-decoder structure is reminiscent of Autoencoder (Hinton & Salakhutdinov, 2006).

In other words, our neural network structure is the conventional GCN concatenated with a Transposed Graph Convolutional Network, which is used to reconstruct the original node features. The loss function of the proposed framework consists of two parts: (1) the supervised loss w.r.t. the labeled part of the graph and (2) the unsupervised reconstruction errors of features, i.e., the regularizer. The SRGCN architecture has several desirable properties for few-shot learning tasks including:

1. The regularization term $\mathcal{L}_{sr}$ is parameterized by a graph convolutional network, whose structure depends on the graph topology (encoded in $g_{\theta_2}$) with a trainable $\theta_2$, and thus has much more expressive powerful than standard regularizations.
2. The reconstruction term in the loss function effectively prevents critical feature information from being washed out during the layer-wise averaging process and could potentially be a building block for training deeper GCNs.
3. In the original GCN, the training of weight matrices is mainly guided by the supervised loss, which is less efficient for the purpose of dimensionality reduction and denoising especially when the training set is small and the feature dimensionality is high, while the Autoencoder in SRGCN allows the neural network to learn the real signals more efficiently.
4. It is twice deeper than GCN, and thus good for label information propagation, which is particular helpful in the few-shot setting.

We assess our new framework on three widely used benchmarks: *Cora, Citeseer* and *Pubmed*. The experimental results show that the proposed approach can significantly improve generalization when there are very few examples from each class, compared with strong baselines such as GCN, GAT (Velickovic et al., 2018), MoNet Monti et al. (2017) etc. For completeness, we also evaluate our model on standard dataset splits, in which the label rate is relatively high. The results show that, with the reconstruction regularization, the accuracy of GCN could be improved in this setting as well. Although GAT achieves slightly better accuracy on *standard splits* than SRGCN, it is arguably much more complicated and does not work well in the few-shot setting.

## 2 SRGCN ARCHITECTURE

### 2.1 PROBLEM DEFINITION

A graph with features is denoted as $G = (V, E, X)$ ,where $V = \{v_1, v_2, v_3, ..., v_n\}$ is the vertex set, $E = \{e_{i,j}\}_{i,j=1}^n$ is the edge set and $X = \{x_1, x_2, ..., x_n\}$ is the set of feature vectors of all nodes. We assume there are totally $c$ classes and each node belongs to exactly one class. The (class) labels of a subset of nodes is revealed in the beginning (i.e., training set), whose indices is denoted by $\mathcal{Y}_L \subseteq \{1, \cdots, n\}$. For each $\ell \in \mathcal{Y}_L$, the label of $v_\ell$ is indicated by a one-hot vector $Y_\ell \in \{0, 1\}^c$. The task is to predict the labels of other nodes. We are most interested in the setting when $|\mathcal{Y}_L|$ is very few and the labels in the training set may not cover all classes.

### 2.2 FRAMEWORK

In this paper, we propose a semi-supervised model named SRGCN. The architecture conceptually consists of two neural networks concatenated together. The first one is a GCN based encoder, while the second one is a decoder, which is essentially a transposed graph convolutional network.

#### 2.2.1 ENCODER

We first describe the encoder part of SRGCN, which is a standard GCN of Kipf & Welling (2017). In general, it is composed of $m_e$ graph convolution layers. The input of each layer is a set of node features represented as a matrix $H^l = \{h_1, h_2, ..., h_n\} \in \mathbb{R}^{n \times d_l}$, in which $n$ is the number of nodes and $d_l$ is the output dimension of the previous layer. The output of the layer is a new set of node features, denoted as $H^{l+1} = \{h'_1, h'_2, ..., h'_n\} \in \mathbb{R}^{n \times d_{l+1}}$, where typically $d_{l+1} < d_l$. More precisely, the layer-wise propagation rule applied on the $l$-th layer can be formalized as follows:

$$H^{l+1} = \sigma(\tilde{D}^{-\frac{1}{2}}\tilde{A}\tilde{D}^{-\frac{1}{2}}H^lW^l).$$

Here, $\tilde{A} = A + I_n$ is the adjacency matrix of the graph after adding self-loops to each node, where $I_n$ is the identity matrix; $\tilde{D}$ is a diagonal matrix with $\tilde{D}_{ii} = \sum_j \tilde{A}_{ij}$; $W^l \in \mathbb{R}^{d \times k}$ is the trainable weight matrix of layer $l$; and $\sigma()$ is an activation function (usually ReLU or softmax). In the 0-th layer we have $H^0 = X$, i.e. using original features as input.

The output of the last layer, denoted as $Z \in \mathbb{R}^{d \times c}$, is the matrix containing the probabilities that every node $v$ belongs to different classes. Here, we used $\theta$ to denote the set of all trainable parameters and the supervised loss is the cross-entropy error, which is defined as follows:

$$\mathcal{L}_{sup}(\theta; A, X) = -\sum_{\ell \in \mathcal{Y}_L} \sum_{j=1}^c Y_{\ell j} \ln Z_{\ell j}.$$

#### 2.2.2 DECODER

For notational convenience, we define $\hat{A} = \tilde{D}^{-\frac{1}{2}}\tilde{A}\tilde{D}^{-\frac{1}{2}}$, which is symmetric. After the dimensionality of the feature space is reduced with a graph convolutional network, a series of transposed graph convolutional layers, each of which is of the form $H' = \sigma(\hat{A}^T H W^T) = \sigma(\hat{A}H W^T)$, are applied to reconstruct the original features of all nodes from their low dimensional representations. Here $W \in \mathbb{R}^{d' \times d}$ is a trainable weight matrix with $d' > d$, so that the dimensionality of feature vectors increases after each layer. Moreover, as mentioned earlier, iterative use of the smoothing operator $\hat{A}X$ results in losing some critical feature information, we hence add a pooling operation (usually max or mean) in each layer to prevent over-smoothing as in (Hamilton et al., 2017; Xu et al., 2018).

In all, the decoder part of the SRGCN consists of the $m_d$ layers of transposed graph convolutions. Each layer takes in as input a set of hidden features, $H^l = \{h_1, h_2, ..., h_n\} \in \mathbb{R}^{n \times d_l}$ and produces a new set of node features $H^{l+1} = \{h'_1, h'_2, ..., h'_n\} \in \mathbb{R}^{n \times d_{l+1}}$, where typically $d_{l+1} > d_l$. More specifically, the layer-wise propagation rule can be written as follows:

$$H^{l+1} = \sigma(\text{pooling}(\tilde{D}^{-\frac{1}{2}}\tilde{A}\tilde{D}^{-\frac{1}{2}}H^l, H^l)W^l)$$

The transposed graph convolutional network use $H^0 = Z$ as the input, whose last layer outputs $\hat{X}$, a matrix of the same size as $X$. The reconstruction loss function is the squared Euclidean distance (or squared Frobenius norm of matrices):

$$\mathcal{L}_{sr}(\theta; A, X) = \| \hat{X} - X \|^2$$

## 2.3 IMPLEMENTATION DETAILS

For all experiments in this paper, SRGCN uses the following four-layer network architecture:

$$
\begin{cases}
H^1 = \mathsf{ReLU}(\hat{A}XW^1), W^1 \in R^{d_0 \times d_1} \\
Z = \mathsf{softmax}(f(\hat{A}H^1W^2)), W^2 \in R^{d_1 \times c} \\
H^2 = \mathsf{ReLU}(\max(Z, \hat{A}Z)W^3), W^3 \in R^{c \times d_1} \\
\hat{X} = \mathsf{ReLU}(\max(H^2, \hat{A}H^2)W^4), W^4 \in R^{d_1 \times d_0}
\end{cases}
$$

Here, $c$ is the number of classes and the pooling method is simply an entry-wise max function. This max-pooling operation does not introduce any additional parameters. Both GCN and GAT use a $L_2$ regularizer in all the experiments in (Kipf & Welling, 2017) and (Velickovic et al., 2018), which is also included in our loss function. Thus, the actual loss function of SRGCN in all experiments is: $\mathcal{L} = \mathcal{L}_{sup} + \alpha \mathcal{L}_{sr} + \beta L_2$. Note that, without the $\mathcal{L}_{sr}$ term, our model is essentially the same as L-2 regularized GCN.

## 3 LIMITATIONS OF SHALLOW GCNs

In this section we discuss the limitations of shallow GCN (refer to the encoder part of SRGCN for the definition of GCN). Suppose the number of layers in GCN is $K$ and let $H^K$ be the output of the last layer. We first assume the loss is a function of the final representations of the labeled nodes, denoted as $H^K_{\mathcal{Y}_L}$ and their corresponding labels $Y_{\mathcal{Y}_L}$, i.e., $\mathcal{L} = f(H^K_{\mathcal{Y}_L}, Y_{\mathcal{Y}_L})$, which is typically the cross entropy function.

Let $N_i(x)$ be the order $i$ neighborhood of node $x$, i.e., the set of nodes connected to $x$ by a path of length at most $i$. For any set $S \subseteq V$, we define $N_i(S) = \bigcup_{x \in S} N_i(x)$. By a simple induction argument, we have the following lemma.

**Lemma 1.** $H^K_{\mathcal{Y}_L}$ *does not depend on the input features of nodes* $v \in V \setminus N_K(\mathcal{Y}_L)$*, and is only a function of the weight matrices* $W^1, \cdots, W^{K-1}$ *and the feature vectors of nodes in the set* $N_K(\mathcal{Y}_L)$ *denoted as* $X_{N_K(\mathcal{Y}_L)}$*.*

Therefore
$$\mathcal{L} = f(H^K_{\mathcal{Y}_L}, Y_{\mathcal{Y}_L}) = g(W^1, \cdots, W^{K-1}, X_{N_K(\mathcal{Y}_L)}, Y_{\mathcal{Y}_L}) \tag{1}$$
for some function $g$. In the classification problem, $g$, $X_{N_K(\mathcal{Y}_L)}$ and $Y_{\mathcal{Y}_L}$ are given and the training algorithm is to find the optimal weight matrices to minimize the loss function. Let $W^*$ be the optimal parameters computed by the algorithm. If we assume the optimization algorithm only uses the information of the loss function to update the optimization variables in each iteration, e.g., its gradients or the Hessian matrix w.r.t. $W^1, \cdots, W^{K-1}$, then $W^*$ must be independent of the feature vectors of nodes $v \in V \setminus N_K(\mathcal{Y}_L)$.

**Lemma 2.** $W^*$ *is function of* $X_{N_K(\mathcal{Y}_L)}, Y_{\mathcal{Y}_L}$*. Any changes in the input features* $X_{V \setminus N_K(\mathcal{Y}_L)}$ *will not affect the value of* $W^*$ *as long as the random seeds used in the optimization algorithm is fixed.*

The above Lemma shows that in the training of parameters, $X_{N_K(\mathcal{Y}_L)}, Y_{\mathcal{Y}_L}$ is the only relevant input data. When the labeled set $\mathcal{Y}_L$ is very small and the depth of the GCN $K$ is small ($K = 2$ is commonly used), the set $N_K(\mathcal{Y}_L)$ only consists of a negligible fraction of the entire graph, and thus most of the input features of the graph are ignored by the training algorithm.

Moreover, applying a standard regularization does not bring any benefits. To see this, if we add a penalty term $r(W)$ that only depends on the parameters $W$, e.g., $L_1$ or $L_2$ regularization, the overall loss is still a function of $W^1, \cdots, W^{K-1}, X_{N_K(\mathcal{Y}_L)}, Y_{\mathcal{Y}_L}$ (from equation 1). Hence the conclusion of Lemma 2 still holds.

On the other hand, our regularization is $\mathcal{L}_{sr} =\parallel \hat{X} - X \parallel^2$, where $X$ includes the input features of all nodes. This force the training algorithm to utilize the information of the entire graph, not just the order $K$ neighborhood of the labeled set.

# 4 RELATED WORK

Our approach extends graph convolutional networks of Kipf & Welling (2017), and thus can be categorized as a spectral approach (Bruna et al., 2014; Henaff et al., 2015; Defferrard et al., 2016). Such spectral methods generalize convolutions to the graph domain, working with a spectral operator depending on the graph structure, which learns hidden layer representations that encode both graph structure and node features simultaneously. Our network structure follows the work of Kipf & Welling (2017), which simplifies previous spectral techniques by restricting the propagation to a 1-hop neighborhood in each layer. Chen et al. (2018) propose fast GCNs, which improves the training speed of the original GCN. GAT of Velickovic et al. (2018) allows for assigning different importances to nodes of a same neighborhood via attention mechanisms. Xu et al. (2018) introduce JK networks, which adaptively adjust (i.e., learn) the influence radii for each node and task. Zügner et al. (2018) study adversarial attacks on neural networks for graphs. In contrast to this paper, such work does not explicitly consider the few-shot learning problem and their performance often degenerates rapidly as the number of labeled samples decreases.

Another direction that generalizes convolutions to the graph structured data, namely non-spectral approaches, define convolutions directly on the graph (Duvenaud et al., 2015; Atwood & Towsley, 2016; Monti et al., 2017). Such methods are easier to be adapted to do inductive learning Hamilton et al. (2017); Velickovic et al. (2018); Bojchevski & Günnemann (2018). However, few-shot learning remains a challenge for this class of methods.

On the other hand, node classification is also one of the main applications of network embedding methods, which learns a lower-dimensional representation for each node in an unsupervised manner, followed by a supervised classifier layer for node classification (Perozzi et al., 2014; Tang et al., 2015; Grover & Leskovec, 2016; Wang et al., 2016; Bojchevski & Günnemann, 2018). DeepWalk of Perozzi et al. (2014) uses local information obtained from truncated random walks to learn latent representations. It assumes that a pair of nodes are similar if they are close in the random walks. LINE preserves the first-order and second order proximity between nodes respectively, and concatenates the representations for the first-order and second order proximity (Tang et al., 2015). A recent work of Bojchevski & Günnemann (2018) proposes Graph2Gauss. This method embeds each node as a Gaussian distribution according to a novel ranking similarity based on the shortest path between nodes. A distribution embedding naturally captures the uncertainty about the representation. The work of (Zhou et al., 2018) tackles the challenge of rare category characterization, which is a similar but different problem considered in this paper. Embedding approaches achieve comparable performance in node classification tasks, while the learned representations also prove to be extremely helpful for other downstream applications.

# 5 EXPERIMENT

In this section, extensive empirical results on three commonly used datasets are provided. In most previous work, a standard split of each datasets[1] is used to evaluate accuracies. Since this work focuses on the few-shot setting, we evaluate the performance of each method using training sets of varying size that are much smaller than in the standard split. The training samples in our experiments are randomly selected from the graph. We observe that, when the training set is extremely small, the performance of each method is highly sensitive to which subset of nodes are labeled. Therefore, in each experiment (with fixed sample size), we evaluate each method on $50$ random splits and report the average accuracy, standard deviation, and a detailed accuracy distribution.

Moreover, the results of SRGCN under standard splits are also presented. Under this setting, SRGCN shows a clear improvement over the original GCN, which demonstrates the effectiveness of our regularization technique. The results of GAT are slightly better than those of SRGCN. GAT

---

[1]The training set contains 20 labeled nodes from each class.

Table 1: Summary of the datasets used in our experiments.

|                     | Cora | Citeseer | Pubmed |
|---------------------|------|----------|--------|
| # of Nodes          | 2708 | 3327     | 19717  |
| # of Edges          | 5429 | 4732     | 44338  |
| # of Features/node  | 1433 | 3703     | 500    |
| # of Classes        | 7    | 6        | 3      |

uses sophisticated attention mechanisms and achieve state-of-the-art accuracies over standard splits, but is more prone to overfitting in few-shot learning.

## 5.1 EXPERIMENT SETTING

We evaluated the performance of SRGCN in two classification tasks. (1) In the first task, the set of labeled nodes (i.e., training set) is uniformly randomly sampled from the graph, so the number of samples from each class is random. Moreover, when the size of the training set is small relative to the number of classes, it is likely that some type of labels may not appear in the training set, i.e., zero-shot learning. (2) In the second task, the training set contains the same number of nodes from each class; and the samples from each class are randomly selected from the class. In the above two tasks, the validation set is chosen in the same size and manner as the training set, and the rest of nodes will be used as the testing set.

**Baseline Methods:** We compare SRGCN with six representative baseline methods: DeepWalk (Perozzi et al., 2014), LINE (Tang et al., 2015), Graph2Gauss (Bojchevski & Günnemann, 2018), GCN (Kipf & Welling, 2017), MoNet(Monti et al., 2017) and GAT (Velickovic et al., 2018). DeepWalk, LINE and Graph2Gauss are unsupervised network embedding methods. For node classification, after the embeddings of nodes are learned, a classifier is trained by applying logistic regression in the embedding space. GCN, MoNet and GAT are semi-supervised node classification methods.

**Datasets.** We consider three most commonly used citation network datasets: Cora, Citeseer and Pubmed. Each node in the graph corresponds to a document and the edges are citation links between documents; the feature vector of a node is a sparse bag-of-words vector. See Table 1 for more details.

**Tasks.** For task 1, we evaluate the classification performance with respect to five different training set sizes: $10, 20, 30, 40, 50$. In this task, zero-shot instances are likely to occur. Note that the logistic regression step in embedding-based methods is unable to deal with zero-shot instances. Hence, we only use GCN, MoNet and GAT as baselines in this task. For task 2, we test the performance with varying numbers of labeled examples from each class (more specifically $1, 3, 5, 7, 10$). In this task, SRGCN are compared with both embedding-based and semi-supervised methods. For each particular training set size, we evaluate each method on 50 random splits and report the average accuracy, standard deviation, and a detailed accuracy distribution.

**Parameter Settings.** We train SRGCN for a maximum of 300 epochs and use the Adam optimizer with learning rate $0.01$. To better assess the effectiveness of our SRGCN, we use the same set of hyperparameters as in (Kipf & Welling, 2017) for GCN. More precisely, the dropout rate is $0.5$, the number of the hidden units in $H_1$ and $H_2$ is 16 and the weight of $L_2$ regularization is $\beta = 0.0005$. Hence the only difference between GCN and SRGCN in the actual implementations is the reconstruction loss. In all the experiments, the weight of the reconstruction loss is set to $\alpha = 0.0001$, which is the only one extra hyperparameter needs to be tuned compared with GCN.

The parameter settings of MoNet and GAT are directly taken from (Monti et al., 2017) and (Velickovic et al., 2018). The embedding size of each unsupervised learning method is set to 128 and other parameters are set in common with settings in (Bojchevski & Günnemann, 2018).

## 5.2 EXPERIMENT RESULTS

The empirical results of task 1 and task 2 are summarized in Table 2 and Table 3 respectively. We complement our few-shot results with experiments on standard splits, the results of which are presented in Table 4.

Table 2: Experimental results on task 1. The numbers in the first row indicate varying sizes of training set, the nodes in which are randomly sampled from the entire graph.

| | Cora | | | | |
|---|---|---|---|---|---|
| | 10 | 20 | 30 | 40 | 50 |
| GCN | 44.69±8.62 | 57.84±7.89 | 65.42±5.86 | 67.74±4.56 | 72.64±3.47 |
| MoNet | 43.92±8.61 | 56.20±7.48 | 62.08±5.35 | 65.43±4.08 | 70.04±3.55 |
| GAT | 35.78±12.17 | 50.45±12.35 | 59.58±8.33 | 62.35±5.24 | 67.72±4.25 |
| SRGCN | **50.04**±11.73 | **64.18**±7.11 | **70.13**±4.25 | **71.63**±4.03 | **76.00**±2.64 |
| | Citeseer | | | | |
| | 10 | 20 | 30 | 40 | 50 |
| GCN | 37.14±8.37 | 46.19±6.67 | 55.28±6.37 | 58.40±4.12 | 60.77±4.25 |
| MoNet | 37.58±7.02 | 45.36±7.61 | 54.43±6.79 | 57.22±5.59 | 59.94±4.78 |
| GAT | 30.69±8.67 | 36.04±11.63 | 50.88±8.02 | 51.14±6.86 | 54.10±4.74 |
| SRGCN | **48.84**±10.76 | **57.99**±7.09 | **64.04**±6.47 | **66.60**±3.00 | **67.72**±2.17 |
| | Pubmed | | | | |
| | 10 | 20 | 30 | 40 | 50 |
| GCN | 56.88±9.17 | 66.13±6.03 | 71.67±5.27 | 74.22±3.05 | **76.64**±2.76 |
| MoNet | 55.57±7.67 | 64.03±5.53 | 68.38±4.22 | 69.64±2.86 | 70.81±2.17 |
| GAT | 46.27±9.11 | 55.14±7.55 | 63.41±5.43 | 66.20±4.05 | 68.41±3.59 |
| SRGCN | **56.98**±9.36 | **66.39**±5.68 | **72.42**±4.50 | **74.33**±3.02 | 76.54±2.84 |

For task 1, the empirical results clearly demonstrate the advantage of SRGCN across all three datasets. In particular, SRGCN achieves significant improvements over GCN (by a margin of up to 6% and 10% on Cora and Citeseer, respectively), while GCN consistently outperforms the other two baselines. Since GCN and SRGCN use the same encoding network structure and the same set of hyperparameters, the effectiveness of the proposed SR regularization is obvious from the results. Perhaps surprisingly, we observe that the performance of GAT is much worse than other baselines in the few-show setting, even though it achieves state-of-the-art accuracy for standard splits. We think a possible reason for this might be that the attention mechanism in GAT results in higher model complexity, which requires more training samples and thus is more likely to cause overfitting in few-show learning. Our regularization method might be a solution to this issue of GAT, which is left as future work.

We observe the classification accuracy is sensitive to the selection of labeled examples in the few-shot setting. In our experiments, the results w.r.t. 50 random splits are recorded. From the standard deviations, we see that SRGCN is generally more robust against the choice of training samples than the baselines. We present box-plots of the accuracy distributions on Cora for task 1 in Figure 1. More results on accuracy distributions can be found in Appendix B.

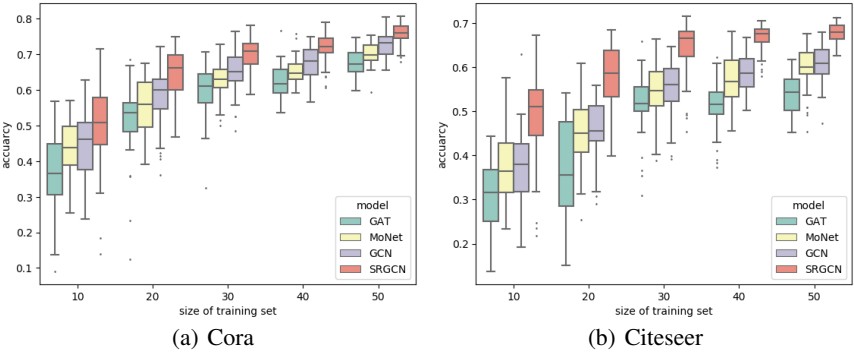

(a) Cora      (b) Citeseer

Figure 1: Box-plot of the accuracies of all models on Cora and Citeseer.

In task 2, compared with all embedding-based methods and semi-supervised methods, SRGCN achieves the highest accuracy under almost all experimental settings. We observe that the recent G2G method has impressive performance especially in one-shot experiments. For GAT, contrast to task 1, when training examples are evenly distributed among all classes, it attains very

competitive performance. Among three semi-supervised baselines, MoNet typically has the best performance. On the other hand, as the number of labeled examples grows, the accuracy of GCN, MoNet and GAT become closer to SRGCN. This is reasonable, since the effect of structural regularization will be dominated by the supervised loss when the training set is large.

Table 3: Experimental results on task 2. The numbers in the first row indicate the number of training samples in each class.

| | Cora | | | | |
|---|---|---|---|---|---|
| | 1 | 3 | 5 | 7 | 10 |
| DeepWalk | 40.42±6.55 | 53.87±4.79 | 59.40±3.77 | 62.25±3.17 | 65.44±3.29 |
| LINE | 49.48±8.70 | 62.68±4.39 | 63.41±6.74 | 69.00±3.11 | 71.13±1.84 |
| G2G | **54.56**±10.34 | 68.12±5.52 | 70.93±2.69 | 72.24±2.08 | 73.89±1.68 |
| GCN | 43.29±8.42 | 63.32±6.26 | 69.33±4.39 | 72.30±3.56 | 74.64±3.20 |
| MoNet | 43.49±7.63 | 61.22±6.27 | 70.94±2.38 | 72.85±2.86 | 76.19±1.96 |
| GAT | 41.83±7.07 | 61.70±7.21 | 71.17±3.49 | 73.46±3.95 | 76.02±2.06 |
| SRGCN | 52.14±10.96 | **68.22**±5.45 | **72.84**±3.48 | **74.79**±2.96 | **76.60**±2.23 |

| | Citeseer | | | | |
|---|---|---|---|---|---|
| | 1 | 3 | 5 | 7 | 10 |
| DeepWalk | 28.38±6.07 | 34.72±4.53 | 38.18±3.02 | 39.76 ±3.32 | 42.01±2.92 |
| LINE | 28.08±5.36 | 34.70±4.74 | 38.03±3.18 | 40.63±2.81 | 43.19±2.55 |
| G2G | **45.10**±9.58 | 56.46±5.45 | 60.38±2.91 | 62.91±2.44 | 63.15±2.00 |
| GCN | 34.94±7.57 | 51.27±6.07 | 57.63±4.04 | 60.89±3.50 | 62.84±2.37 |
| MoNet | 38.80±6.77 | 52.95±5.96 | 59.79±3.91 | 62.93±3.00 | 64.69±2.72 |
| GAT | 32.88±9.49 | 48.68±5.93 | 54.92±4.95 | 57.91±4.68 | 60.83±4.74 |
| SRGCN | 45.05±11.29 | **59.32**±7.24 | **64.35**±3.82 | **66.39**±2.46 | **67.03**±2.23 |

Table 4: Experimental results on standard splits.

| | Cora | Citeseer | Pubmed |
|---|---|---|---|
| GCN | 81.4±0.5 | 70.9±0.5 | **79.0**±0.3 |
| MoNet | 81.7±0.5 | — | 78.8±0.3 |
| GAT | **83.0**±0.7 | **72.5**±0.7 | **79.0**±0.3 |
| SRGCN | 82.3±0.6 | 71.8±0.4 | **79.0**±0.3 |

Finally, we report the classification accuracies of semi-supervised models on the standard dataset splits, which are commonly used in previous works such as Yang et al. (2016); Kipf & Welling (2017); Velickovic et al. (2018). Results for all baseline methods are taken from the original papers. Even in this setting, where the label rate is relatively high [2], SRGCN still outperforms GCN. This shows that our SR regularization not only improves accuracy significantly for few-shot learning, the results on standard splits shows that it is also an effective regularization method for general purpose. Equipped with attention mechanisms, GAT remains the best for standard splits. Since our regularization technique is quite general, it is natural to ask whether it could be applied on GAT or other graph neural networks to improve their generalization, which is left as future work.

## 6 CONCLUSION

We have introduced a novel regularized GCN architecture for few-shot classification problems on graph-structured data. Our SRGCN model combines the original GCN structure with an encoder-decoder architecture, which can be considered as a regularization method. What make this more powerful than standard methods is that the regularization term is a function parameterized by a trainable neural network, whose structure depends on the underlying graph topology. Experiments on a number of network datasets shows that SRGCN has matched state-of-the-art results and can significantly improve classification accuracy when there are very few examples from each class. Moreover, as the original GCN, our method is simple and easy to implement. This suggest that the proposed SRGCN model is capable of overcoming the classification accuracy degeneration caused by insufficient labeled information, while being computationally efficient on par with GCNs.

---

[2]In the standard splits, the number of labels per class is 20

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

## A    NETWORK VISUALIZATION

Semi-supervised methods also learn a low-dimensional embedding for each node in the graph, which can be used to create meaningful visualizations of a network in 2D/3D. Moreover, the effectiveness of such methods can also be investigated qualitatively when the learned representations are further projected onto the 2D space. We provide such a visualization of the feature representation produced by GCN and SRGCN respectively on the Cora dataset (Figure 2) with t-SNE (Maaten & Hinton, 2008). The visualization results show that the clustering effect exhibited by SRGCN is more discernible than that by GCN, which align with our numerical results.

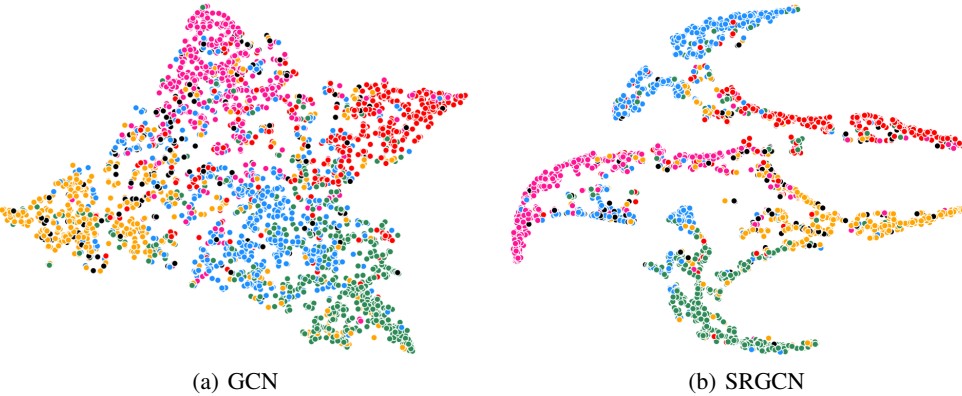

(a) GCN                                    (b) SRGCN

Figure 2: Visualization: t-SNE plots of the computed feature representations of GCN's last layer and SRGCN's middle layer (used for classification) on the Citeseer dataset. The representations are trained using a labeled set of size 40.

## B    MORE EXPERIMENTAL RESULTS

We present the histograms of the accuracy distributions on Cora and Citeseer about task 1. As is shown in the histograms, X-axis represents accuracy, Y-axis represents the frequency of the experiment whose accuracy is in this interval.

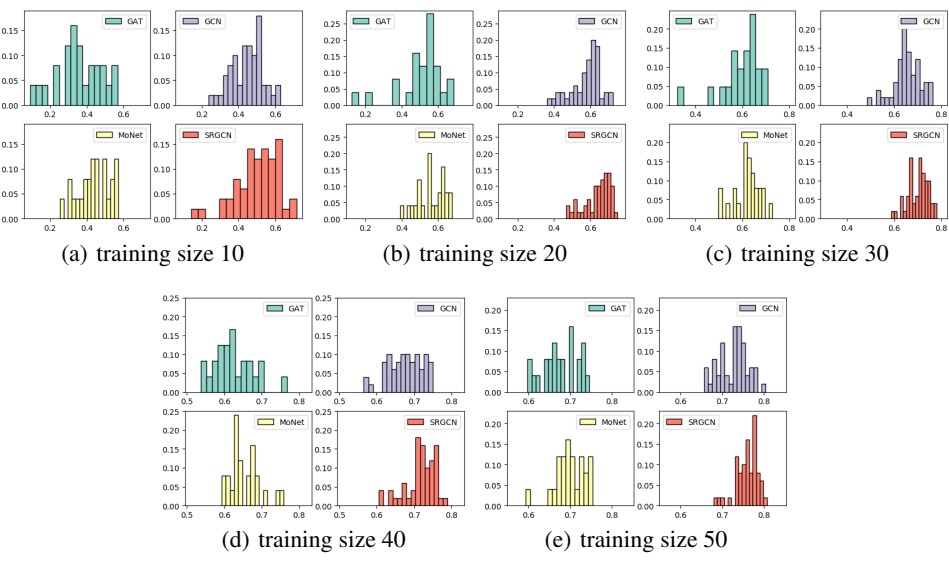

Figure 3: Histograms of task 1 for each model on Cora.

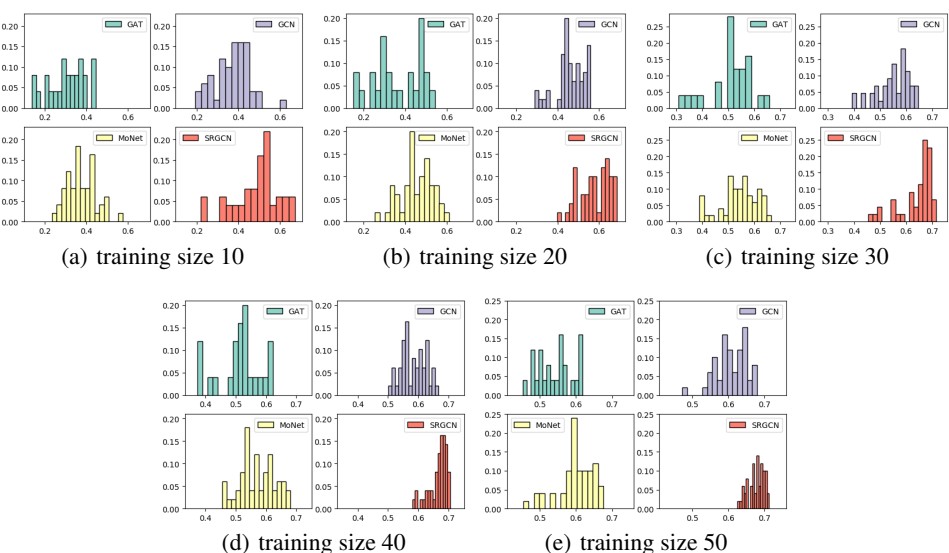

Figure 4: Histograms of task 1 for each model on Citeseer.

