# OpenReview forum: "Few-shot Classification on Graphs with Structural Regularized GCNs"
_ICLR.cc/2019/Conference_

### Official Review · AnonReviewer3 · 2018-10-31
**Idea is reasonable; work is preliminary**

**Rating:** 5
**Confidence:** 4

**Review:**

Edited: I raised the score by 1 point after the authors revised the paper significantly.

--------------------------------------------

This paper proposes a regularization approach for improving GCN when the training examples are very few. The regularization is the reconstruction loss of the node features under an autoencoder. The encoder is the usual GCN whereas the decoder is a transpose version of it.

The approach is reasonable because the unsupervised loss restrains GCN from being overfitted with very few unknown labels. However, this paper appears to be rushed in the last minute and more work is needed before it reaches an acceptable level.

1. Theorem 1 is dubious and the proof is not mathematical. The result is derived based on the ignorance of the nonlinearities of the network. The authors hide the assumption of linearity in the proof rather than stating it in the theorem. Moreover, the justification of why activation functions can be ignored is handwavy and not mathematical.

2. In Section 2.2 the authors write "... framework is shown in Figure X" without even showing the figure.

3. The current experimental results may be strengthened, based on Figures 1 and 2, through showing the accuracy distribution of GAT as well and thoroughly discussing the results.

4. There are numerous grammatical errors throughout the paper. Casual reading catches these typos: "vertices which satisfies", "makes W be affected", "the some strong baseline methods", "a set research papers", and "in align with". The authors are suggested to do a thorough proofreading.

---

> ### Author Response · Authors · 2018-11-26
> **Response to Reviewer3**
>
> We thank the reviewer for the constructive comments and have revised the paper significantly.
> 1) We have greatly simplified the mathematics in section 3. We reorganized the material and simplified the notations that causing much confusion before. In particular, we removed the notion of influence distribution of Xu et al., as we find that this notion is not necessary for our purpose. We think that the revised version provides a clearer and more mathematical explanation on why shallow GCN is not sufficient for few-shot learning and why standard regularization doesn’t help.
>
> 2) We have made a thorough revision and proofreading to eliminate grammatical errors throughout the paper. We are sorry for causing much trouble in the first version.
>
> 3) In order to strengthen the experimental results, we revised the experimental part. We have added MoNet (suggested by Reviewer 2) as another baseline and provide standard deviation of accuracy in all the experimental results as suggested. Some of the results are as follows, and the rest is in the revised paper.
> -----------------------------------------------------------------------------------------------------------------
>                                                                        Cora
> -----------------------------------------------------------------------------------------------------------------
>                            10                        20                       30                        40                      50
> -----------------------------------------------------------------------------------------------------------------
> GCN       44.69 +/- 8.62    57.84 +/- 7.89    65.42 +/- 5.86    67.74 +/- 4.56   72.64 +/- 3.47
> -----------------------------------------------------------------------------------------------------------------
> MoNet  43.92 +/- 8.61    56.20 +/- 7.48    62.08 +/- 5.35    65.43 +/- 4.08   70.04 +/- 3.55
> -----------------------------------------------------------------------------------------------------------------
> GAT      35.78 +/- 12.17  50.45 +/-12.35   59.58 +/- 8.33    62.35 +/- 5.24    67.72 +/- 4.25
> -----------------------------------------------------------------------------------------------------------------
> SRGCN 50.04 +/- 11.73  64.18 +/- 7.11    70.13 +/- 4.25     71.63 +/- 4.03   76.00 +/- 2.64
> -----------------------------------------------------------------------------------------------------------------
> In addition, we provide the accuracy distribution of GAT and MoNet. Their box plots are shown in Section 5.2 and distribution histograms are presented in the appendix due space constraint. Furthermore, we have provided more detailed discussions on the experimental results.
>
> 4) We also add results on standard splits. In this setting, SRGCN is still better than GCN and MoNet and is only slightly inferior to GAT, while being arguably much simpler than GAT. The results are as follows (the experimental results except SRGCN are copied from previous work) .
> ----------------------------------------------------------------------------
>                       Cora              Citeseer            Pubmed
> -----------------------------------------------------------------------------
> GCN        81.4 +/- 0.5     70.9 +/- 0.5        79.0 +/- 0.3
> -----------------------------------------------------------------------------
> MoNet    81.7 +/- 0.5             ---                78.8 +/- 0.3
> -----------------------------------------------------------------------------
> GAT         83.0 +/- 0.7     72.5 +/- 0.7        79.0 +/- 0.3
> -----------------------------------------------------------------------------
> SRGCN    82.3 +/- 0.6     71.8 +/- 0.4        79.0 +/- 0.3
> -----------------------------------------------------------------------------
>
> These new experiments confirm our previous claims that our SR regularization could improve accuracy significantly for few-shot learning. The results on standard splits shows that it is also an effective regularization method for general purpose.

---

> > ### Comment · AnonReviewer3 · 2018-11-27
> > **Paper is improved**
> >
> > I appreciate the authors' efforts on revising the paper.
> >
> > Now the theory reads more convincing than before. It makes sense to use an autoencoder to restrict the parameterization through considering additionally the nodes not occurring in the supervised loss.
> >
> > Experiments are much enriched.
> >
> > I raised the score by 1 point. (There might be a line regarding how much change can be made in ICLR; I somehow feel that this paper has been edited too much, understandably because of the poor quality of the initial submission.)
> >
> > The theory could be strengthened if section 3 is expanded with empirical illustrations of the node coverage under only the supervised loss and a shallow GCN architecture.
> >
> > Still, the writing could be further improved.

---

### Official Review · AnonReviewer2 · 2018-11-01
**interesting extension to GCNs, somehwat lacking a comprehensive evaluation**

**Rating:** 6
**Confidence:** 3

**Review:**

I appreciate the author response and additional effort to provide comparison with MoNet. I have raised my rating by 1 point. It should be noted that the edits to the revision are quite substantial and more in line of a journal revision. My understanding is that only moderate changes to the initial submission are acceptable.

-----------------------------------------------

The paper introduces a new regularization approach for graph convolutional networks. A transposed GCN is appended to a regular GCN, resulting in a trainable, graph specific regularization term modelled as an additional neural network.

Experiments demonstrate performance en par with previous work in the case where sufficient labelled data is available. The SRGCNs seem to shine when only few labelled data is available (few shot setting).

The method is appealing as the regularization adapts to the underlying graph structure, unlike structure-agnostic regularization such as L1.

Unclear why the results are not compared to MoNet (Monti et al. 2017) which seems to be the current state-of-the-art for semi-supervised classification of graph nodes.

Overall, well written paper with an interesting extension to GCN. The paper is lacking a comprehensive evaluation and comparison to latest work on graph neural networks. The results in the few shot setting are compelling.

---

> ### Author Response · Authors · 2018-11-26
> **Response to Reviewer2**
>
> Thanks for the constructive comments and feedback. We have revised the paper accordingly. Major revisions are summarized as follows.
> 1) We have added MoNet as another baseline as suggested. We have also added the standard deviation of accuracy in all the experimental results as suggested by reviewer 3. The experimental results of MoNet on task 1 on Cora and Citeseer are as follows (see our revised paper for more results on MoNet)
> -----------------------------------------------------------------------------------------------------------------
>                                                                         Cora
> -----------------------------------------------------------------------------------------------------------------
>                           10                        20                       30                      40                       50
> -----------------------------------------------------------------------------------------------------------------
> MoNet  43.92 +/- 8.61    56.20 +/- 7.48   62.08 +/- 5.35   65.43 +/- 4.08   70.04 +/- 3.55
> -----------------------------------------------------------------------------------------------------------------
> SRGCN 50.04 +/- 11.73   64.18 +/- 7.11  70.13 +/- 4.25   71.63 +/- 4.03   76.00 +/- 2.64
> -----------------------------------------------------------------------------------------------------------------
> ----------------------------------------------------------------------------------------------------------------
>                                                                      Citeseer
> -----------------------------------------------------------------------------------------------------------------
>                             10                        20                        30                         40                     50
> -----------------------------------------------------------------------------------------------------------------
> MoNet   37.58 +/- 7.02      45.36 +/- 7.61    54.43 +/- 6.79    57.22 +/- 5.59    59.94 +/- 4.78
> -----------------------------------------------------------------------------------------------------------------
> SRGCN  48.84 +/-10.76      57.99 +/- 7.09    64.04 +/- 6.47    66.60 +/- 3.00    67.72 +/- 2.17
> -----------------------------------------------------------------------------------------------------------------
> From the results, our SRGCN also outperforms MoNet by a large margin. Compared with other baselines, MoNet is generally worse than GCN in task 1 but is more competitive in task 2.
>
> 2) We also add results on standard splits. In this setting, SRGCN is still better than GCN and MoNet and is only slightly inferior to GAT, while being arguably much simpler than GAT. The results are as follows (the experimental results except SRGCN are copied from previous work) .
> ----------------------------------------------------------------------------
>                       Cora                Citeseer                Pubmed
> -----------------------------------------------------------------------------
> GCN       81.4 +/- 0.5         70.9 +/- 0.5          79.0 +/- 0.3
> -----------------------------------------------------------------------------
> MoNet   81.7 +/- 0.5                 ---                  78.8 +/- 0.3
> -----------------------------------------------------------------------------
> GAT        83.0 +/- 0.7         72.5 +/- 0.7          79.0 +/- 0.3
> -----------------------------------------------------------------------------
> SRGCN   82.3 +/- 0.6         71.8 +/- 0.4          79.0 +/- 0.3
> -----------------------------------------------------------------------------
> These new experiments confirm our previous claims that our SR regularization could improve accuracy significantly for few-shot learning. The results on standard splits shows that it is also an effective regularization method for general purpose.
>
> 3) We have greatly simplified the mathematics in section 3. We reorganized the material and simplified the notations that causing much confusion before. In particular, we removed the notion of influence distribution of Xu et al., as we find that this notion is not necessary for our purpose. We think that the revised version provides a clearer and more mathematical explanation on why shallow GCN is not sufficient for few-shot learning and why standard regularization doesn’t help.

---

### Official Review · AnonReviewer1 · 2018-11-02
**presentation could be significantly improved, details are missing, validation is not compelling**

**Rating:** 4
**Confidence:** 4

**Review:**

This paper proposes to regularize the training of graph convolutional neural networks by adding a reconstruction loss to the supervised loss. Results are reported on citation benchmarks and compared for increasing number of labeled data.

The presentation of the paper could be significantly improved. Details of the proposed model are missing and the effects of the proposed regularization w.r.t. other regularizations are not analyzed.

My main concerns are related to the model design, the novelty of the approach (adding a reconstruction loss) and its experimental evaluation.

Details / references of the transposed convolution operation are missing (see e.g. https://ieeexplore.ieee.org/document/7742951). It is not clear what the role of the transposed convolution is in that case. It seems that the encoder does not change the nodes nor the edges of the graph, only the features, and the filters of the transposed convolution are learnt. If the operation is analogous to the transposed convolution on images, then given that the number of nodes in the graph does not change in the encoder layers (no graph coarsening operations are applied), then learning an additional convolution should be analogous (see e.g. https://arxiv.org/pdf/1603.07285.pdf Figure 4.3.). Could the authors comment on that?

Details on the pooling operation performed after the transposed convolution are missing (see e.g. https://arxiv.org/pdf/1805.00165.pdf, https://arxiv.org/pdf/1606.09375.pdf). Does the pooling operation coarsen the graph? if so, how is it then upsampled to match the input graph?

Figure X in section 2.1. does no exist.

Supervised loss in section 2.2.1 seems to disregard the sum over the nodes which have labels.

\hat A is not defined when it is introduced (in section 2.2.2), it appears later in section 2.3.

Section 2.2.2 suggests that additional regularization (such as L2) is still required (note that the introduction outlines the proposed loss as a combination of reconstruction loss and supervised loss). An ablation study using either one of both regularizers should be performed to better understand their impact. Note that hyper-parameters chosen give higher weight to L2 regularizer.

Section 3 introduces a bunch of definitions to presumably compare GCN against SRGCN, but those measures of influence are not reported for any model.

Experimental validation raises some concerns. It is not clear whether standard splits for the reported datasets are used. It is not clear whether hyper-parameter tuning has been performed for baselines. Authors state "the parameters of GCN and GAT and SRGCN are the same following (Kipf et al; Velickovic et al.)". Note that SRGCN probably has additional parameters, due to the decoder stacked on top of the GCN. Reporting the number of parameters that each model has would provide more insights. Results are not reported following standards of running the models N times and providing mean and std. Moreover, there are no results using the full training set.

---

> ### Author Response · Authors · 2018-11-26
> **Response to Reviewer1 (Part 1/2)**
>
> We thank the reviewer for the comments and we have revised the paper thoroughly. We are sorry for causing confusion and misunderstandings on some technical issues. We make the following clarifications.
> 1) As noted by the reviewer, convolution operator in GCN is actually quite different from convolution on images. The encoder does not change the nodes nor the edges of the graph, only the features. In particular the dimensionality of the output feature space is much lower.
>
> 2)##comment on transposed graph convolution##
> Our main idea is to reconstruct the original features and use the reconstruction errors as a regularization, so the output feature vectors in the low-dimensional space needs to be transformed back into the original space; in particular, the dimensionality needs to be lifted to match the original space. Therefore, one can think of the transposed GCN as a GCN being reversed, which also does not change the graph, but only the features.
> To illustrate this, let’s consider a simple one-layer GCN as an example. Now the output of GCN is Z=\sigma(\hat{A} XW), where X is the input feature matrix and W is a trainable weigh matrix. To reconstruct X from Z, the transposed GCN is applied on Z: X’=\sigma(\hat{A}^T Z W’^T). Here \hat{A} is symmetric, so the transpose operator could be omitted and W’ is another trainable weight matrix. To make the dimensionality of X’ and X the same, W and W’ must have the same size and the linear transformation W’ also needs to be transposed before applying on Z.
>
> 3) ##Clarification on pooling##
> The pooling method used here follows e.g. https://arxiv.org/pdf/1806.03536.pdf, https://arxiv.org/abs/1706.02216, which is different from pooling in CNN for images. Here, we simply use the entry-wise max function: max(X, \hat{A}X), which also does not change the graph, but only the features. So, it is more like an activation function and doesn’t coarsen the graph. We followed previous work and called this pooling. We are sorry for causing confusion. Now we have made the pooling more explicit in the revised paper and hope that this resolves the concern of the reviewer.
>
> 4) In our original paper, the supervised loss in section 2.2.1 is ambiguous. So, we have revised it in the new version.
>
> 5) We have added the definition \hat A in section 2.2.2 as suggested by the reviewer.
>
> 6) ##comment on L2 regularization##
> Both GCN and GAT use a L2 regularizer in all their experiments. But as demonstrated by our empirical and theoretical results (section 3 in the new version), L2 regularizer is not enough to handle few-show learning. Our main contribution is a new regularization. The experiments show that the performance of GCN + L2 + new regularizer significantly outperforms GCN + L2. So we think it is fair enough to say that our new regularizer is highly effective.
> We emphasize that we don’t mean to replace standard regularization methods.
>
> 7) ##Simplified analysis##
> We have greatly simplified the mathematics in section 3. We reorganized the material and simplified the notations that causing much confusion before. In particular, we removed the notion of influence distribution of Xu et al., as we find that this notion is not necessary for our purpose. We think that the revised version provides a clearer and more mathematical explanation on why shallow GCN is not sufficient for few-shot learning and why standard regularization doesn’t help.
>
> 8) ##comment on higher weight for L2 regularizer##
> For the weights of the regularizers, one should see that higher weight doesn’t mean the corresponding regularizer is more important. In our case, although a higher weight is given to the L2 regularizer compared with the reconstruction loss (0.0005 vs 0.0001), the overall effect of the reconstruction loss is much stronger. The main reason that the weight of the reconstruction regularizer is smaller is that the reconstruction loss is a function of all feature vectors, whose total size is n*d (n is the number of nodes and d is the number of input features). n*d is typically much larger than the number of parameters in our setting. In particular, after training, 0.0001*reconstruction loss is always much larger than 0.0005*L2.
>
> 9) ##Hyperparameters##
> Note the encoder and decoder of SRGCN are two symmetric GCNs and we use the same set of hyper-parameters for them, which are exactly the same as suggested by Kipf et al. The only additional parameter SRGCN has is the weight of the reconstruction loss, which is set to 0.0001 in all experiments. All the parameters were explicitly listed in section 5.1.

---

> > ### Author Response · Authors · 2018-11-26
> > **Response to Reviewer1 (Part 2/2)**
> >
> > 10) ##Clarification on data splits##
> > In the standard splits, the label rate is relatively high, so for few-shot experiments, we use random splits. For each training size, we test each model on 50 random splits and report the average accuracy and the accuracy distribution. In the revision, we also report standard deviations as suggested. In order to strengthen the experimental results, we revised the experimental part. We have added MoNet as another baseline. Some of the results are as follows; and see the revised paper for more results.
> > -----------------------------------------------------------------------------------------------------------------
> >                                                                        Cora
> > -----------------------------------------------------------------------------------------------------------------
> >                            10                        20                       30                        40                      50
> > -----------------------------------------------------------------------------------------------------------------
> > GCN       44.69 +/- 8.62    57.84 +/- 7.89    65.42 +/- 5.86    67.74 +/- 4.56   72.64 +/- 3.47
> > -----------------------------------------------------------------------------------------------------------------
> > MoNet  43.92 +/- 8.61    56.20 +/- 7.48    62.08 +/- 5.35    65.43 +/- 4.08   70.04 +/- 3.55
> > -----------------------------------------------------------------------------------------------------------------
> > GAT      35.78 +/- 12.17  50.45 +/-12.35   59.58 +/- 8.33    62.35 +/- 5.24    67.72 +/- 4.25
> > -----------------------------------------------------------------------------------------------------------------
> > SRGCN 50.04 +/- 11.73  64.18 +/- 7.11    70.13 +/- 4.25     71.63 +/- 4.03   76.00 +/- 2.64
> > -----------------------------------------------------------------------------------------------------------------
> >
> > 11)##Results on standard splits##
> > In the revision, we also report experimental results on standard splits. In this setting, SRGCN is still better than GCN and MoNet and is only slightly inferior to GAT, while being arguably much simpler than GAT. The results are as follows (the experimental results except SRGCN are copied from previous work).
> > ----------------------------------------------------------------------------
> >                       Cora                Citeseer                Pubmed
> > -----------------------------------------------------------------------------
> > GCN       81.4 +/- 0.5         70.9 +/- 0.5          79.0 +/- 0.3
> > -----------------------------------------------------------------------------
> > MoNet   81.7 +/- 0.5                 ---                  78.8 +/- 0.3
> > -----------------------------------------------------------------------------
> > GAT        83.0 +/- 0.7         72.5 +/- 0.7          79.0 +/- 0.3
> > -----------------------------------------------------------------------------
> > SRGCN   82.3 +/- 0.6         71.8 +/- 0.4          79.0 +/- 0.3
> > -----------------------------------------------------------------------------
> > These new experiments confirm our previous claims that our SR regularization could improve accuracy significantly for few-shot learning. The results on standard splits shows that it is also an effective regularization method for general purpose.

---

### Meta-Review · Area_Chair1 · 2018-12-17
**interesting extension; but too preliminary**

**Confidence:** 5
**Recommendation:** Reject

**Metareview:**

A new regularized graph CNN approach is proposed for semi-supervised learning on graphs.  The conventional Graph CNN is concatenated with a Transposed Network, which is used to supplement the supervised loss w.r.t. the labeled part of the graph with an unsupervised loss that serves as a regularizer measuring reconstruction errors of features. While this extension performs well and was found to be interesting in general by the reviewers,  the novelty of the approach (adding a reconstruction loss),  the completeness of the experimental evaluation, and the presentation quality have also been questioned consistently. The paper has improved during the course of the review, but overall the AC evaluates that paper is not upto ICLR-2019 standards in its current form.